# *GPR143*-Associated Ocular Albinism in a Hispanic Family and Review of the Literature

**DOI:** 10.3390/genes16080911

**Published:** 2025-07-30

**Authors:** Anushree Aneja, Brenda L. Bohnsack, Valerie Allegretti, Allison Goetsch Weisman, Andy Drackley, Alexander Ing, Patrick McMullen, Andrew Skol, Hantamalala Ralay Ranaivo, Kai Lee Yap, Pamela Rathbun, Adam Gordon, Jennifer L. Rossen

**Affiliations:** 1Department of Ophthalmology, Feinberg School of Medicine, Northwestern University, Chicago, IL 60611, USA; anushree.aneja@northwestern.edu (A.A.); bbohnsack@luriechildrens.org (B.L.B.); aweisman@luriechildrens.org (A.G.W.); 2Division of Ophthalmology, Ann & Robert H. Lurie Children’s Hospital of Chicago, Chicago, IL 60611, USA; vallegretti@luriechildrens.org (V.A.); adrackley@luriechildrens.org (A.D.); aing@luriechildrens.org (A.I.); pmcmullen@luriechildrens.org (P.M.); askol@luriechildrens.org (A.S.); hralay@luriechildrens.org (H.R.R.); klyap@luriechildrens.org (K.L.Y.); prathbun@luriechildrens.org (P.R.); 3Center for Genetic Medicine, Northwestern University, Chicago, IL 60611, USA; adam.gordon@northwestern.edu

**Keywords:** ocular albinism, *GPR143*, nystagmus, foveal hypoplasia

## Abstract

**Background/Objectives:** While ocular albinism (OA) is usually associated with reduced vision, nystagmus, and foveal hypoplasia, there is phenotypic variability in iris and fundus hypopigmentation. Hemizygous pathogenic/likely pathogenic (P/LP) variants in *GPR143* at X: 151.56–151.59 have been shown in the literature to be associated with OA. The purpose of this study was to report the case of a Hispanic male with X-linked inherited OA associated with a hemizygous *GPR143* variant and to review the literature relating to genotype–phenotype associations with *GPR143* and OA. **Methods:** After consent to an IRB-approved protocol, a 14-year-old Hispanic male patient with OA and his parents underwent whole genome sequencing (WGS) in 2023. Two maternal uncles with nystagmus underwent targeted variant testing in 2024. A literature review of reported *GPR143* variants was completed. **Results:** A male with reduced visual acuity, infantile-onset nystagmus, foveal hypoplasia, and iris hypopigmentation was identified to have the variant *GPR143*, c.455+3A>G, which was also present in his mother and two affected maternal uncles. This variant has been previously identified in other Hispanic patients of Mexican descent. Additionally, 127 variants were identified in the literature and reported to be associated with OA. All patients had reduced visual acuity (average 0.71 ± 0.23 logMAR), 99% had nystagmus, 97% foveal hypoplasia, 79% fundus hypopigmentation, and 71% iris hypopigmentation. Of those patients with reported optotype best corrected visual acuity (BCVA), eight (9%) had VA from 20/25 to 20/40, 24 (24%) had VA from 20/50 to 20/80, and 63 (67%) had VA from 20/100 to 20/200. The most frequent type of variant was missense (31%, n = 39). Frameshift and nonsense variants were associated with the lowest rates of iris hypopigmentation (50% [n = 11] and 44% [n = 8], respectively; *p* = 0.0068). **Conclusions:** This case represents phenotypic variability of *GPR143*-associated OA and highlights the importance of repeat genetic testing and independent analyses of test results for accurate variant classification, particularly in non-White and Hispanic patients. Further studies in more diverse populations are needed to better develop genotype–phenotype associations for *GPR143*-associated OA.

## 1. Introduction

Ocular albinism type 1 (OA1) is a condition of X-linked inheritance characterized by broad phenotypic ocular variability generally without dermatologic manifestations. It is believed to occur in 1 in 60,000 males [1]. Ocular manifestations may include moderately to severely reduced visual acuity, nystagmus, strabismus, photophobia, iris translucency/transillumination defects (TIDs), hypopigmented fundi, and foveal hypoplasia. X-linked hemizygous variants in *GPR143* gene are a well-known genetic cause for OA1 [2]. However, since the phenotype is variable, particularly with often mild or even absent hypopigmentation of the iris and fundus, and no dermatologic manifestations of albinism, the diagnosis may be easily missed. While the disease is not progressive, due to the stable reduced visual acuity, patients with ocular albinism often require low vision services. Additionally, patients with ocular albinism may be at an increased risk of neurodevelopmental impairment, requiring early intervention services [3].

The *GPR143* gene, located on the X chromosome, spans 40 kB of genomic DNA and consists of nine coding exons, creating a protein that is 404 amino acids in length [4]. It encodes a G-protein-coupled receptor expressed on the apical aspect of retinal pigment epithelial cells that plays a critical role in melanosome biogenesis [4]. Pathogenic variants in *GPR143* disrupt this receptor’s function, resulting in the enlargement and misshaping of melanosomes, termed as “macromelanosomes”, which impairs melanin synthesis [5]. Melanin plays a key role in cell polarity and foveal development; thus, the disruption of this pathway leads to a decrease in pigment production within the eye and symptoms consistent with OA, including hypopigmented iris and fundus with foveal hypoplasia and iris transillumination defects [1,6].

We present a case of a diagnosis of *GPR143*-associated OA in an adolescent Hispanic male with infantile-onset nystagmus with prior negative genetic testing during infancy and a significant family history of nystagmus in maternal male relatives. Whole exome sequencing was completed as a trio with biological parents, and familial variant analysis was completed for two affected maternal uncles. We also review and analyze *GPR143*-associated OA genotype–phenotype variant associations reported in the literature.

## 2. Materials and Methods

Written informed consent for a prospective research protocol was obtained from the proband’s parents under an IRB-approved study (IRB 2021-4730), and the study abided the tenets of the Declaration of Helsinki and was conducted in accordance with the Health Insurance Portability and Accountability Act.

The proband and his parents received pre- and post-test genetic counseling with a genetic counselor and trio-based whole genome sequencing (WGS) at a CLIA-certified commercial laboratory in 2023. Their proprietary analysis pipeline was utilized with reads also aligned to genome build GRCh37/hg19, citing that 98.9% of targeted regions were covered at a minimum depth of 10x, an overall average depth of 143×, and CNV resolution at a level of approximately three or more exons. The trio WES dataset was re-analyzed by the Lurie Children’s Molecular Diagnostics Laboratory on a research basis, using the Illumina DRAGEN Bio-IT Platform 3.9 with alignment of the data to genome build GRCh38/hg38. The proband’s maternal uncles underwent targeted variant testing at the same commercial laboratory to evaluate for the *GPR143* variant previously identified in the proband and mother.

A literature review was also performed to describe previously reported *GPR143* variants with phenotypic associations. Papers were identified initially through review of the Human Gene Mutation Database (HGMD) which was queried to identify publications with phenotype data on *GPR143* variants. Additional papers were identified from the references of the papers identified from the HGMD. Papers were included if they provided ophthalmologic findings for patients with reported *GPR143* variants. Demographics and clinical findings were obtained along with molecular genetic findings. Best corrected visual acuity (BCVA) was documented as the best visual acuity reported for either eye or both eyes together. BCVA, if reported as optotype, was converted to logMAR scale. Many, but not all, papers included at least some description of ocular manifestations: visual acuity, nystagmus, foveal hypoplasia, fundus hypopigmentation, and iris hypopigmentation. Genotype/phenotype information was combined to evaluate the prevalence of different clinical findings as well as the location, type, and frequency of specific variants. Statistical analysis was completed with Fischer exact test.

## 3. Results

A newborn Hispanic male of Mexican descent was referred to the Division of Ophthalmology at the Ann & Robert H. Lurie Children’s Hospital of Chicago for congenital nystagmus in 2009. He had a significant maternal history of infantile-onset nystagmus. While his mother did not have nystagmus, his maternal uncles had nystagmus (Figure 1). The patient’s father and three sisters did not have nystagmus; he did not have any brothers. His ophthalmic exam was significant for blunted foveal reflex and mild iris TIDs without cutaneous or hair hypopigmentation. Due to concern for possible ocular albinism, he underwent genetic testing in 2009 at the age of 6 months-old by a commercial laboratory for a variant in *GPR143*. The testing was conducted using Sanger sequencing of *GPR143* including only coding regions and the immediately flanking intron sequences. The test report indicated no identified variants.

In 2023, at age 14 years, the patient presented for regular ophthalmologic follow-up. His exam was notable for nystagmus with very minimal TIDs. His best-corrected visual acuity was 20/70 in each eye and binocularly. Clinical exam and optical coherence tomography of the macula showed blunted foveal reflexes, prominent choroidal vessels and lack of foveal contour consistent with fundus hypopigmentation and foveal hypoplasia (Figure 2). The patient and his family were interested in further genetic testing to identify the cause of the infantile-onset nystagmus that was prevalent in their family.

Starting in 2022, our multidisciplinary ophthalmic genetics research team began offering patients with undiagnosed likely genetic ophthalmic conditions trio whole exome sequencing (WES) or WGS in a CLIA-certified commercial laboratory. In 2023, after being consented and undergoing testing by the commercial laboratory, the patient was identified to have a maternally inherited variant in *GPR143*, NM_000273.3: c.455+3A>G that was classified as variant of uncertain significance (VUS).

To evaluate for segregation of this variant with disease in the family, two of the proband’s maternal uncles with nystagmus and reduced visual acuity were offered targeted familial variant testing as standard of care follow-up testing after results were obtained for the proband and his parents, performed by the commercial laboratory that originally performed the WGS analysis. Both uncles were identified to be hemizygous for this variant as well; the variant was still classified and reported as a VUS. Evaluation of the variant by members of the Lurie Children’s Molecular Diagnostics Laboratory later resulted in their classification of the variant as pathogenic (ACMG-AMP criteria applied: PVS1 [RNA], PS4_moderate, PP1_moderate) [7,8].

### Literature Review

A total of 40 papers were identified containing genetic variants (n = 127) in *GPR143* associated with OA1 with reports on patients from the United States (10 families), Canada (6 families) Africa (2 families), Europe (43 families, mostly western Europe) Asia (59 families from China, 6 from Korea and 1 from Japan), and Mexico (2 families) [2,4,5,9,10,11,12,13,14,15,16,17,18,19,20,21,22,23,24,25,26,27,28,29,30,31,32,33,34,35,36,37,38,39,40,41,42,43,44,45,46,47]. Race/ethnicity were not always described in the reports with only 3 previous reports of Hispanic families. The variants and clinical findings were collated in Appendix A. Many, but not all, publications included variant-specific data for the ocular manifestations including reduced vision (which varied in definition by each literature report but was always below 20/20)), nystagmus, foveal hypoplasia, fundus hypopigmentation, and iris hypopigmentation. A total of 167 patients had at least some ocular phenotypic data reported. When combining the results of the literature reports, 100% of patients (149 of 149) had reduced vision, 99% (165 of 167) had nystagmus, 97% (113 of 117) had foveal hypoplasia, 79% (91 of 115) had fundus hypopigmentation and 71% (93 of 115) had iris hypopigmentation (Table 1). There was data on optotype visual acuity for 94 patients, ranging from 0.1 to 1.0 logMAR) with an average of 0.71 ± 0.23 [IQR 0.6, 0.9]. Eight patients (9%) had VA from 20/25 to 20/40, 24 patients (24%) had VA from 20/50 to 20/80, and 63 patients (67%) had VA from 20/100 to 20/200.

The variants span the entire gene (Figure 3). There were five variants reported at least three times in the literature (Table 2). All families reported hemizygous affected males, and one consanguineous family also had homozygous affected females [31]. The most common types of variants were missense (31%, n = 39), followed by large deletions (26%, n = 33) and frameshift (20%, n = 25) (Table 3). The only clinical finding that was statistically significantly different across variant type was iris hypopigmentation (Table 4, *p*-value 0.0068), which was least prevalent in nonsense (44%, n = 8) and frameshift (50%, n = 11) variants.

## 4. Discussion

Hemizygous variants in *GPR143* include various forms: missense, nonsense, splice site, frameshift, small insertions or deletions, and gross deletions of some or all exons. Our patient’s variant, c.455+3A>G, is in the splice donor site of intron 3, conferring significant impact on transcription and subsequent protein assembly. Analysis utilizing SpliceAI generated strong computational predictions of a splicing impact (SpliceAI donor loss Δ = 0.75) [48] specifically that loss of recognition of this donor splice site would likely cause the preceding exon to be skipped, causing a shift to the mRNA reading frame, and likely triggering nonsense-mediated decay with a resultant loss of *GPR143*. Indeed, Sepúlveda-Vázquez et al. [24] demonstrated in patient-derived RNA that this variant results in a complete loss of *GPR143* expression, consistent with its predicted effect to splicing and the resultant mRNA transcript.

While our patient was identified to have a splicing variant on WGS but not on initial targeted sequencing of the coding regions of *GPR143*, splicing variants are the least common variants reported in *GPR143*, making up 16% of variants. The most common variants identified in the literature are missense (31%), large deletions (26%), and frameshift (20%). Splice variants reported in the literature have multiple subsequent effects on the *GPR143* protein, including the creation of new acceptor and donor splice sites and introduction of premature stop codons, thus truncating the protein and inhibiting interaction with crucial downstream signaling molecules that regulate melanosome production and function [49]. Missense variants frequently arise within the second and third cytosolic loops of the protein and can result in aberrant protein folding and decreased intrinsic signaling protein function [33]. Additionally, one patient was reported to have a synonymous variant, c.360G>A (p.Ala120=), however it alters the last nucleotide of the exon within the consensus splice donor site of exon 2 and had been graded as likely pathogenic on ClinVar [47].

Our patient had numerous significant clinical findings on examination consistent with phenotypic characteristics outlined in the literature, for which there is a broad spectrum. His visual acuity (20/70) was slightly better than the average (20/100), and he displayed the characteristic phenotype of infantile-onset nystagmus, foveal hypoplasia, iris TIDs and fundus hypopigmentation. In our literature review, all patients with *GPR143*-associated OA and ocular phenotypic data (n = 158) have been reported to have reduced visual acuity (the definition varies by publication), but it ranges from 20/25 to 20/200, with about 2/3 of patients having VA equal or worse than 20/100. The second most common finding was nystagmus (99%) followed by foveal hypoplasia (97%). In general, iris and fundus hypopigmentation were rarer, 71% and 79%, respectively, but these findings may be more subtle and missed especially in uncooperative children.

In comparing the prevalence of ocular clinical manifestations by variant type, only the presence or absence of iris hypopigmentation was statistically significantly different (*p*-value 0.0105). Patients with frameshift and nonsense variants appear to have lower rates of iris pigmentation (50% and 44%, respectively), compared to large deletions (83%), missense (82%), and splicing (75%) variants. While not statistically significant, frameshift variants had the lowest rate of fundus hypopigmentation (64%) compared to other variant types (*p*-value 0.3542). This difference in frequency of clinical characteristics according to variant type, while only present for one symptom, underscores the need for expansive genetic testing to better phenotypically characterize these variants to facilitate more targeted genetic testing to sooner identify the variant specific to the patient. Race has also been linked to variation in clinical findings. While it has been suggested that white patients are more likely to have iris hypopigmentation/TIDs than Asian and Black patients [11], in our literature review, 98% (115 of 117) of Asian patients had iris hypopigmentation, which was actually higher than the 71% reported rate in all patients. And there were very few Hispanic and Black patients previously reported. Findings of infantile-onset nystagmus with foveal hypoplasia (with or without iris and fundus hypopigmentation) along with a family history suggestive of an X-linked inheritance pattern should warrant genetic testing including analysis of coding and non-coding regions of *GPR143*.

Additional associations, such as neurodevelopment impairment, have also been reported in patients with OA secondary to *GPR143* variants [3]. However, none were noted for our patient or were a focus of other reports. It is important for future reports to document on neurodevelopmental issues to better understand associations with *GPR143* variants and neurodevelopment and then aid guidance for parents to consider early interventions when appropriate.

The variant identified in our patient and his family was initially classified by the commercial laboratory as a VUS. Our variant classification experts (AD and AI) subsequently did a reanalysis of the variant NM_000273.3 (*GPR143*): c.455+3A>G and classified it as pathogenic, supporting this patient’s diagnosis of OA (ACMG-AMP criteria applied: PS3, PS4_moderate, PP4). This classification discordance appears to be due to differences in the application of ACMG-AMP evidence criteria rather than availability of new data. This same variant has been previously identified in the hemizygous state in several unrelated Hispanic males from Mexico with OA, and at least once was shown to be inherited from a patient’s mildly affected mother [24,47]. We previously reported an unrelated Hispanic male of Mexican descent with infantile-onset nystagmus and fundus hypopigmentation, without cutaneous or hair manifestations nor iris TIDs, who was hemizygous for the same c.455_3A>G variant (15). Furthermore, this is the only variant to be described previously in Hispanic populations, although previous literature has been minimal in this ethnic group. The co-segregation of this variant with features of OA in the family presented in this report, including our patient’s maternal uncles, adds further support of the variant’s role in disease for this and other families, and is sufficient to consider this variant as pathogenic (ACMG-AMP criteria applied: PVS1 [RNA], PS4_moderate, PP1_moderate) [7,8]. The discrepancies in classifications of this variant underscore the importance of independent review of testing restuls and ClinGen variant curation expert panels to reduce inconsistencies in the application of the ACMG-AMP criteria.

As shown through our review of the literature, the majority of patients diagnosed previously with *GPR143*-associated OA are either Asian or European, so it is possible that the variants more prevalent in patients from other countries and of non-White or non-Asian races/ethnicities, such as in our Hispanic patients, are less well characterized. More testing and scientific reporting is needed in diverse populations to better understand the clinical significance of these variants and to develop genotype–phenotype associations for *GPR143*-related OA.

## 5. Conclusions

In conclusion, hemizygous variants in *GPR143* have been shown in the literature to be strongly associated with infantile-onset nystagmus, reduced vision, and foveal hypoplasia with varying degrees of iris and fundus hypopigmentation. While the majority of previous research has been in patients of European and Asian descent, more studies are needed in those of different racial and ethnic backgrounds to develop a better understanding of genotype–phenotype associations.

## Figures and Tables

**Figure 1 genes-16-00911-f001:**
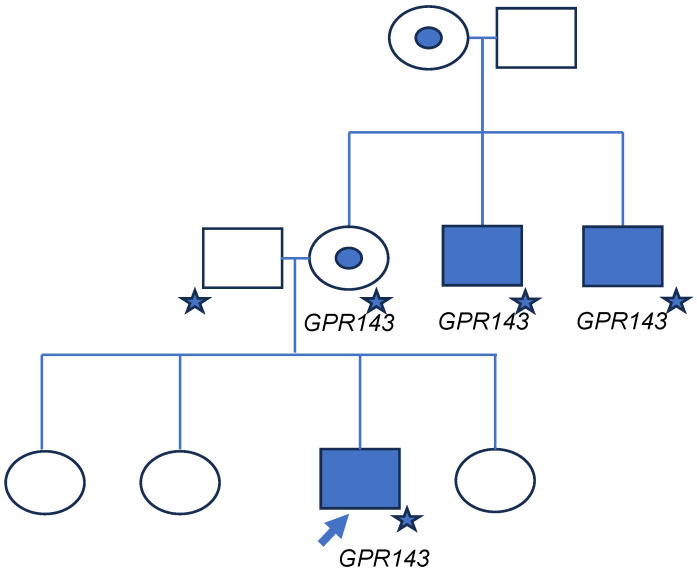
Pedigree. Family pedigree. Filled square indicates individuals affected with low vision and nystagmus. Circles with inner circles indicate carriers (either by genetic testing or obligate carriers). Unfilled circles and squares are unaffected individuals. Arrow indicates proband. Star indicates patient received genetic testing. Those with the suspicious variant in *GPR143* are indicated.

**Figure 2 genes-16-00911-f002:**
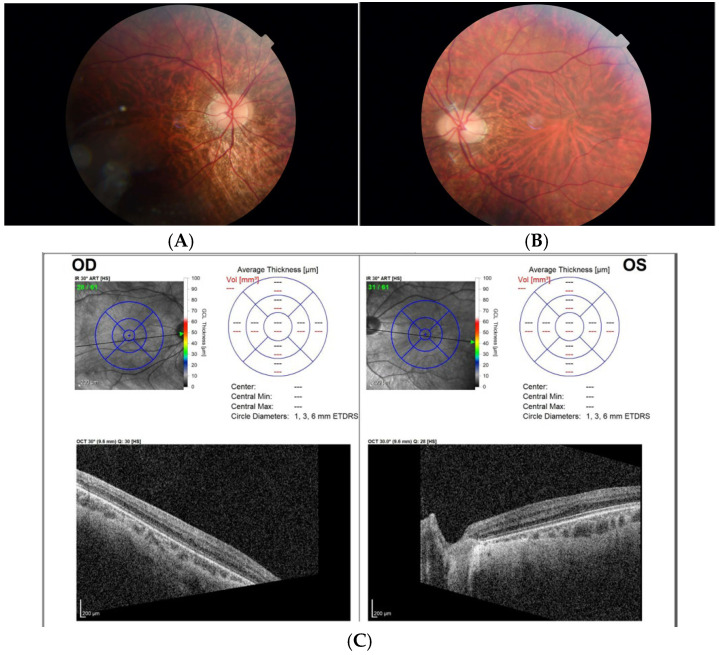
Fundus photography and OCT macula of proband. Fundus photos of the right (**A**) and left (**B**) eyes show blunted foveal reflex and a prominent appearance of the choroidal vessels. OCT macula bilateral (**C**) shows reduced foveal contours, consistent with a diagnosis of foveal hypoplasia bilaterally.

**Figure 3 genes-16-00911-f003:**
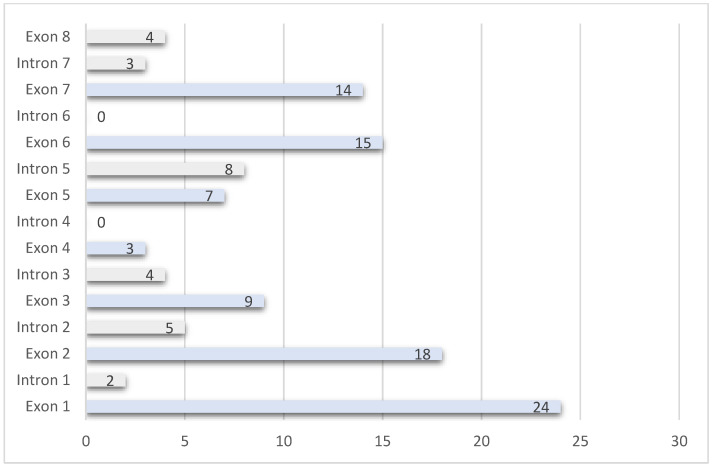
Literature reports of variants in *GPR143* associated with ocular albinism. *X*-axis is the number of variants noted in separate literature reports identified in each exon/intron. The *Y*-axis is the exon/intron number.

**Table 1 genes-16-00911-t001:** Summary of clinical findings in patients with ocular albinism due to variants in *GPR143*.

*Clinical Finding*	Patients with Clinical Finding	Percentage of Patients with Clinical Finding
*Reduced Visual Acuity*	149/149	100%
*Visual Acuity (N = 95 patients)*	0.71 ± 0.23 [IQR 0.6, 0.9].	
*Nystagmus*	165/167	99%
*Foveal Hypoplasia*	113/117	97%
*Fundus Hypopigmentation*	91/115	79%
*Iris Hypopigmentation*	92/130	71%

**Table 2 genes-16-00911-t002:** Variants in *GPR143* associated with ocular albinism reported at least three times in the literature.

*Variant*	# of Literature Reports
*c.251G>A*, *p.Gly84Asp*	3
*c.353G>A*, *p.Gly118Glu*	4
*c.455+3A>G*	3
*c.703G>A*, *p.Glu235Lys*	4
*c.733C>T*, *p.Arg245* *	7

The * symbol refers to a premature stop codon, meaning that at the 245th position in the protein sequence, Arginine has been replaced by a stop codon, leading to a truncated protein. The # symbol refers to the word “Number”, with the title of that column being “Number of Literature Reports”.

**Table 3 genes-16-00911-t003:** Types of variants in *GPR143* associated with ocular albinism reported in the literature.

*Type of Variant*	# of Variants	% of Total Variants (N = 127)
*Missense*	39	31%
*Large Deletion*	33	26%
*Frameshift*	25	20%
*Splicing*	20	16%
*Nonsense*	9	7%
*Synonymous*	1	1%

The # symbol refers to the word “Number”, with the title of that column being “Number of Literature Reports”.

**Table 4 genes-16-00911-t004:** Clinical findings by variant type for variants in *GPR143* associated with ocular albinism reported in the literature.

	Missense	Large Deletion	Frameshift	Splicing	Nonsense	*p*-Value
*Reduced* *Visual Acuity*	100% (47 of 47 patients)	100% (27 of 27 patients)	100% (24 of 24 patients)	100% (31 of 31 patients)	100% (27 of 27 patients)	>0.9999
*Nystagmus*	97% (43 of 44 patients)	100% (40 of 40 patients)	100% (27 of 27 patients)	100% (36 of 36 patients)	100% (19 of 19 patients)	>0.9999
*Iris* *Hypopigmentation*	82% (29 of 35 patients)	83% (19 of 23 patients)	50% (11 of 22 patients)	76% (25 of 33 patients)	44% (8 of 18 patients)	**0.0063**
*Fundus* *Hypopigmentation*	82% (19 of 23 patients)	88% (15 of 17 patients)	64% (14 of 22 patients)	84% (27 of 32 patients)	84% (21 of 25 patients)	0.3228
*Foveal* *Hypoplasia*	95% (22 of 23 patients)	100% (19 of 19 patients)	95% (18 of 19 patients)	100% (31 of 31 patients)	96% (23 of 24 patients)	0.5837

## Data Availability

Data can be seen in the Appendix A.

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
