# Peer review of "GPR143-Associated Ocular Albinism in a Hispanic Family and Review of the Literature"

_genes, 2025, doi:10.3390/genes16080911_

Round 1
Reviewer 1 Report
Comments and Suggestions for Authors
I would like to thank the authors and the editors for the opportunity to review this manuscript, which aims to report the case of a Hispanic male with X-linked inherited ocular albinism (OA) associated with a hemizygous GPR143 variant, and to review the literature on genotype–phenotype associations involving GPR143 and OA. Although the manuscript is of generally good quality, some revisions are necessary before publication. I believe the authors can address the comments successfully and improve the manuscript, enhancing its relevance and appeal to the international scientific community.
It is clear that the authors have focused primarily on the description of the ocular phenotype. However, it would be of interest to expand on other aspects of the patient’s profile, particularly regarding neurodevelopmental aspects. A broader discussion, including a literature review on associated features such as intellectual disability or autism spectrum disorder, would strengthen the manuscript. Notably, a recent study published in 2022 reported cases of individuals with OA associated with GPR143 variants who also presented with neurodevelopmental disorders.
The criteria used for the literature review are not clearly described. It would be important for the authors to specify the time frame considered (i.e., from which year the articles were included), the databases consulted, and the inclusion and exclusion criteria applied. Furthermore, it is unclear whether a structured framework, such as a PICO model, was used to guide the search and selection of relevant studies.
In the Results section, the authors report that the patient has two sisters; however, the family pedigree appears to indicate the presence of three female siblings. This discrepancy should be clarified to ensure consistency between the textual description and the pedigree diagram.
Author Response
I would like to thank the authors and the editors for the opportunity to review this manuscript, which aims to report the case of a Hispanic male with X-linked inherited ocular albinism (OA) associated with a hemizygous GPR143 variant, and to review the literature on genotype–phenotype associations involving GPR143 and OA. Although the manuscript is of generally good quality, some revisions are necessary before publication. I believe the authors can address the comments successfully and improve the manuscript, enhancing its relevance and appeal to the international scientific community.
Comment: It is clear that the authors have focused primarily on the description of the ocular phenotype. However, it would be of interest to expand on other aspects of the patient’s profile, particularly regarding neurodevelopmental aspects. A broader discussion, including a literature review on associated features such as intellectual disability or autism spectrum disorder, would strengthen the manuscript. Notably, a recent study published in 2022 reported cases of individuals with OA associated with GPR143 variants who also presented with neurodevelopmental disorders.
Response: Thank you for your inciteful comment. While most reports do not address neurodevelopmental impairments, we agree it is important to alert readers to this association. In the discussion section, we have included the above reference and commented on the importance of future publications to report on neurodevelopmental issues for patients with GPR143 variants to better understand genotype-phenotype associations.
“Additional associations, such as neurodevelopment impairment, have also been reported in patients with OA secondary to GPR143 variants (49). However, none were noted for our patient or were a focus of other reports. It is important for future reports to document on neurodevelopmental issues to better understand associations with GPR143 variants and neurodevelopment and then guidance for parents to consider early interventions when appropriate.”
Comment: The criteria used for the literature review are not clearly described. It would be important for the authors to specify the time frame considered (i.e., from which year the articles were included), the databases consulted, and the inclusion and exclusion criteria applied. Furthermore, it is unclear whether a structured framework, such as a PICO model, was used to guide the search and selection of relevant studies.
Response: Thank you for this comment. We have included an explanation for the methods of our review in the Materials and Methods section:
“Papers were identified initially through review of the Human Gene Mutation Database (HGMD) which was queried to identify publications with phenotype data on GPR143 variants. Additional papers were identified from the references of the papers identified from the HGMD. Papers were included if they provided ophthalmologic findings for patients with reported GPR143 variants.”
Comment: In the Results section, the authors report that the patient has two sisters; however, the family pedigree appears to indicate the presence of three female siblings. This discrepancy should be clarified to ensure consistency between the textual description and the pedigree diagram.
Response: Thank you for this comment. This has been edited.
Reviewer 2 Report
Comments and Suggestions for Authors
This review article on GPR143-associated ocular albinism (OA) focuses on a Hispanic family and examines existing literature. It highlights the phenotypic variability of the condition and the importance of accurate genetic testing. The goal was to report a case of X-linked inherited OA in a Hispanic male with a hemizygous GPR143 variant, review the current literature on genotype-phenotype relationships for GPR143 and OA, and confirm the genetic basis and variable presentation of the condition. It also stresses the critical need for comprehensive genetic testing and further research in diverse populations to enhance understanding of genotype-phenotype correlations in OA. However, I have mentioned below a few key comments for authors to address.
- The authors have provided a limited number of references to cross-validate and support the statements in this review. The authors have added a more rigorous literature review and incorporated it as needed throughout the manuscript in text citation, for example, tables 1 and 2, and figure 3.
- The details characterization of the Ocular albinism type 1 (OA1) is missing, emphasizing its global prevalence, phenotypes, cause of pathologies, and socioeconomic burden in health care. Authors need to explain and add an appropriate reference.
- In the material and method section authors have not provided any source or citation that can validate the statement for whole genome sequencing, including data. So the authors have justified and included them accordingly.
- In the discussion section, authors needed to add their rational point of view, unlike summarizing the literature, which is likely to address the research gap, and give a novel clue on how to overcome GPR143-Associated Ocular Albinisms-related challenges and limitations, which makes the discussion thought-provoking to the reader and enhances the manuscript.
Author Response
This review article on GPR143-associated ocular albinism (OA) focuses on a Hispanic family and examines existing literature. It highlights the phenotypic variability of the condition and the importance of accurate genetic testing. The goal was to report a case of X-linked inherited OA in a Hispanic male with a hemizygous GPR143 variant, review the current literature on genotype-phenotype relationships for GPR143 and OA, and confirm the genetic basis and variable presentation of the condition. It also stresses the critical need for comprehensive genetic testing and further research in diverse populations to enhance understanding of genotype-phenotype correlations in OA. However, I have mentioned below a few key comments for authors to address.
- Comment: The authors have provided a limited number of references to cross-validate and support the statements in this review. The authors have added a more rigorous literature review and incorporated it as needed throughout the manuscript in text citation, for example, tables 1 and 2, and figure 3.
Response: Thank you for your comment. We have included 40 references with genotype-phenotype associations. These have been included in the manuscript body as well as tables and figures, including the supplementary table that shows the full genotype-phenotype associations.
- Comment: The details characterization of the Ocular albinism type 1 (OA1) is missing, emphasizing its global prevalence, phenotypes, cause of pathologies, and socioeconomic burden in health care. Authors need to explain and add an appropriate reference.
Response: Thank you for your comment. We have added this information to the Introduction.
- Comment: In the material and method section authors have not provided any source or citation that can validate the statement for whole genome sequencing, including data. So the authors have justified and included them accordingly.
Response: Thank you for your comment. We have included the following information on the WGS analysis in the Materials and Methods section:
“The proband and his parents received pre- and post-test genetic counseling with a genetic counselor and trio-based whole genome sequencing (WGS) at a CLIA-certified commercial laboratory in 2023. Their proprietary analysis pipeline was utilized with reads also aligned to genome build GRCh37/hg19, citing that 98.9% of targeted regions were covered at a minimum depth of 10x, an overall average depth of 143x, and CNV resolution at a level of approximately three or more exons. The trio WES dataset was re-analyzed by the Lurie Children’s Molecular Diagnostics Laboratory on a research basis, using the Illumina DRAGEN Bio-IT Platform 3.9 with alignment of the data to genome build GRCh38/hg38.”
- Comment: In the discussion section, authors needed to add their rational point of view, unlike summarizing the literature, which is likely to address the research gap, and give a novel clue on how to overcome GPR143-Associated Ocular Albinisms-related challenges and limitations, which makes the discussion thought-provoking to the reader and enhances the manuscript.
Response: Thank you for your comment. Here are experts from our point of view:
“In general, iris and fundus hypopigmentation were rarer, 71% and 79%, respectively, but these findings may be more subtle and missed especially in uncooperative children.”
“This difference of frequency of clinical characteristics according to variant type, while only present for one symptom, underscores the need for expansive genetic testing to better phenotypically characterize these variants to facilitate more targeted genetic testing to sooner identify the variant specific to the patient. Race has also been linked to variation in clinical findings.”
“Findings of infantile-onset nystagmus with foveal hypoplasia (with or without iris and fundus hypopigmentation) along with a family history suggestive of an X-linked inheritance pattern should warrant genetic testing including analysis of coding and non-coding regions of GPR143. “
“ It is important for future reports to document on neurodevelopmental issues to better understand associations with GPR143 variants and neurodevelopment and then guidance for parents to consider early interventions when appropriate. “
“The discrepancies in classifications of this variant underscore the importance of ClinGen variant curation expert panels to reduce inconsistencies in the application of the ACMG-AMP criteria.”
“As shown through our review of the literature, the majority of patients diagnosed previously with GPR143-associated OA are either Asian or European, so it is possible that the variants more prevalent in patients from other countries and of non-White or non-Asian races/ethnicities, such as in our Hispanic patients, are less well characterized. More testing and scientific reporting is needed in diverse populations to better understand the clinical significance of these variants and to develop genotype-phenotype associations for GPR143-related OA.”
Round 2
Reviewer 1 Report
Comments and Suggestions for Authors
The authors have further improved the paper by becoming more precise in the description of the methodology applied for the literature review and in the description of the clinical symptoms that subjects with the GPR143 mutation may present. In particular, the possibility has been raised that, alongside visual impairments, neurodevelopmental disorders may also be present. I believe this paper can contribute to shedding light on this rare and still poorly described condition.
Reviewer 2 Report
Comments and Suggestions for Authors
Please incorporate and highlight all changes that the authors have made in the revised version of the manuscript. It seems that most of the comments only address in review report, but are not integrated into the revised version of the manuscript.